# The Interplay between T Cells and Cancer: The Basis of Immunotherapy

**DOI:** 10.3390/genes14051008

**Published:** 2023-04-28

**Authors:** Christina Chen, Xin Liu, Che-Yu Chang, Helen Y. Wang, Rong-Fu Wang

**Affiliations:** 1Department of Molecular Microbiology and Immunology, Keck School of Medicine, University of Southern California, Los Angeles, CA 90033, USA; 2Department of Medicine, Keck School of Medicine, University of Southern California, Los Angeles, CA 90033, USA; 3Department of Pediatrics, Children’s Hospital Los Angeles, Keck School of Medicine, University of Southern California, Los Angeles, CA 90033, USA

**Keywords:** cancer, T cells, signaling, alterations, cell therapy, vaccine, antibody, CRISPR, CAR T/TCR T/TIL

## Abstract

Over the past decade, immunotherapy has emerged as one of the most promising approaches to cancer treatment. The use of immune checkpoint inhibitors has resulted in impressive and durable clinical responses in the treatment of various cancers. Additionally, immunotherapy utilizing chimeric antigen receptor (CAR)-engineered T cells has produced robust responses in blood cancers, and T cell receptor (TCR)-engineered T cells are showing promising results in the treatment of solid cancers. Despite these noteworthy advancements in cancer immunotherapy, numerous challenges remain. Some patient populations are unresponsive to immune checkpoint inhibitor therapy, and CAR T cell therapy has yet to show efficacy against solid cancers. In this review, we first discuss the significant role that T cells play in the body’s defense against cancer. We then delve into the mechanisms behind the current challenges facing immunotherapy, starting with T cell exhaustion due to immune checkpoint upregulation and changes in the transcriptional and epigenetic landscapes of dysfunctional T cells. We then discuss cancer-cell-intrinsic characteristics, including molecular alterations in cancer cells and the immunosuppressive nature of the tumor microenvironment (TME), which collectively facilitate tumor cell proliferation, survival, metastasis, and immune evasion. Finally, we examine recent advancements in cancer immunotherapy, with a specific emphasis on T-cell-based treatments.

## 1. Introduction

Cytotoxic T lymphocytes (CTLs), or cytotoxic T cells, play a critical role in the recognition and elimination of tumor cells [1]. Cytotoxic T cells are activated via a two-step process. First, T cell receptors (TCRs) bind to tumor-derived peptides presented on major histocompatibility complex (MHC) molecules by antigen-presenting cells (APCs) [1,2]. This initial interaction establishes the antigen specificity of the T cell response. The second (co-stimulatory) T cell activation signal is mediated by the CD28–B7 interaction [1,2]. The combination of these two signals facilitates the full activation of cytotoxic T cells, enabling them to target and eliminate cells displaying their cognate antigen, including cancer cells, through antigen-specific killing [1,2]. Various cell-based therapies utilize genetically modified cell-surface antigen-specific receptors that aim to enhance and optimize the targeting and cytotoxic activity of T cells [3,4]. For example, TCR T cells and chimeric antigen receptor (CAR) T cells are examples of genetically engineered T cell therapies that have generated durable responses in clinical trials and demonstrate promising antitumor activity [3]. As of 2022, the first United States Food and Drug Administration (FDA)-approved TCR T therapy was granted approval for the treatment of HLA-A*02:01-positive uveal melanoma [5]. Various other TCR T cell clinical trials are currently underway and have shown promising results [3]. CAR T cells have likewise demonstrated impressive antitumor activities. Six CAR T cell products have been approved by the FDA, four of which target CD19-positive tumors and two of which target B cell maturation antigen (BCMA) [4,6,7,8]. A summary of ongoing or completed clinical trials of CAR T and TCR T therapies can be found in Table 1. Although the use of CAR T cells has achieved remarkable success in the treatment of hematological malignancies, poor clinical responses have been observed against solid tumors [4]. Various barriers are currently faced by cancer immunotherapy, including the unique properties of tumor cells that promote growth and immune evasion, the dysfunction of T cells that prevent the effective elimination of tumor cells, and an intricate interplay between tumor cells and T cells in the TME [9]. These obstacles must be overcome to fully harness the potential of cancer immunotherapy [10,11,12,13].

### 1.1. T Cell Receptor Structure and T Cell Signaling

In the early 1980s, studies utilizing immunological and biochemical techniques uncovered the composition and structure of the TCR complex [36,37,38]. The TCR is a heterodimer composed of two chains: α- and β-chains or γ- and δ-chains [39]. In humans, αβ T cells comprise 95% of all T cells and are responsible for recognizing peptides bound to MHC molecules [39]. The remaining 5% of the T cell population is composed of γδ T cells, which have the unique ability to function in MHC-independent antigen recognition [39]. The complete αβ TCR complex requires four additional CD3 polypeptide chains—the epsilon (ε), gamma (γ), delta (δ), and zeta (ζ) chains—that are critical to carrying out signal transduction initiated by TCR ligation [40]. These chains assemble as three pairs of dimers (γε, δε, and ζζ) to form a stable octameric complex with the αβ TCR, creating a complex with equal stoichiometric proportions of TCRαβ, CD3γε, CD3δε, and CD3ζζ [40].

Upon binding to the peptide–MHC complex, the conformation of the TCR complex changes, triggering lymphocyte-specific protein tyrosine kinase (Lck)-mediated phosphorylation of immunoreceptor tyrosine-based activation motifs (ITAMs) within the CD3 adaptor proteins [41]. pITAMs then serve as docking sites for zeta-chain-associated protein kinase 70 (Zap70), which binds via its tandem Src homology 2 (SH2) domains, resulting in its phosphorylation and activation [42,43,44]. Once activated, Zap70 phosphorylates multiple tyrosine residues in the linker for the activation of T cells (LAT), a key scaffold protein that recruits various downstream adaptor molecules and signaling proteins, including the SH-2-domain-containing leukocyte protein of 76 kDa (SLP-76), phospholipase Cγ1 (PLCγ1), son of sevenless (SOS), growth factor receptor-bound protein 2 (GRB2), interleukin (IL)-2-induced tyrosine kinase (ITK), Vav, non-catalytic tyrosine kinase (NCK), adhesion- and degranulation-promoting adapter protein (ADAP), and Grb2-related adaptor downstream of Shc (GADS) [45,46]. These signaling molecules activate the downstream IP_3_-Ca^2+^-NFAT, DAG-PKCθ-NF-κB, and DAG-RAS-AP1 pathways, which work together to promote T cell activation, proliferation, survival, and effector function [42,43,44]. Homeostatic regulation of these cornerstone pathways is critical for normal T cell function. However, various factors within the TME contribute to the dysregulation of these crucial pathways, leading to the impairment of T cell effector functions and cancer immune evasion.

### 1.2. Aberrant T Cell Signaling Associated with Various Malignancies

Dysfunctional T cell signaling has been implicated in multiple malignancies, including head and neck squamous cell carcinoma (HNSCC) [47,48], lung cancer (including non-small-cell lung cancer (NSCLC)) [49,50,51,52], hepatocellular carcinoma (HCC) [47,53,54,55,56,57], gastric cancer [56,58], T cell acute lymphoblastic leukemia (T-ALL) [59], chronic lymphocytic leukemia (CLL) [60,61], melanoma [62,63,64], lymphoma [65,66,67,68,69,70,71,72,73,74,75,76,77,78,79], colon cancer, and others [80]. The causes of these malignancies and their resistance to immunotherapy are multifaceted and can be attributed to altered expression of inhibitory receptors and negative regulators, as well as changes in the transcriptional and epigenetic landscapes of T cells. These changes ultimately result in a dysfunctional phenotype characterized by a loss of T cell effector functions and reduced antitumor efficacy. For instance, reduced interferon-gamma (IFNγ) production is often observed in the tumor-infiltrating lymphocytes (TILs) of patients with different forms of cancer, including melanoma [62], HNSCC [47], and ovarian cancer [81]. Attenuated tumor necrosis factor (TNF) production is seen in T cells isolated from mice with colon carcinoma [80] and in TILs of HNSCC [47] and ovarian cancer [81] patients. High granzyme K levels are observed in dysfunctional CTL populations of melanoma [82], HCC [54], and NSCLC patients [52]. Reduced perforin and granzyme B can also be observed in certain tumor-infiltrating CTLs [62,80].

## 2. Mechanisms Responsible for the Current Challenges Faced in Immunotherapy

### 2.1. Exhaustion and Dysfunction of T Cells

The imperative role of immune protection against cancer highlights the importance of functional T cell signaling and activity in preventing and combating cancer. T cell exhaustion, a hypofunctional T cell state characterized by a loss of self-renewal capacity and reduced effector function, is a significant contributor to cancer progression and constitutes a major obstacle in the successful application of CAR T cell therapy against solid cancers [83,84,85]. The cause of T cell exhaustion in the context of cancer is multifaceted, with persistent antigenic stimulation, the immune-inhibitory nature of the TME, changes in the T-cell-related transcriptional and epigenetic landscapes, and metabolic factors all identified as contributing factors [86].

T cell exhaustion is commonly characterized by an increased expression of inhibitory receptors [84]. Under normal physiological conditions, these inhibitory checkpoints serve a protective function. Ligation of inhibitory receptors (expressed on activated T cells) with their respective ligands (expressed on tissues and APCs) attenuates T cell responses and promotes T cell tolerance, thereby modulating inflammation and preventing collateral damage during chronic infections [87]. However, many tumors take advantage of this protective mechanism by overexpressing inhibitory ligands, which in turn ligate with inhibitory receptors. This triggers intracellular pathways that effectively prevent T cell effector functions against the tumor and consequently results in immune tolerance of the cancer [87].

#### 2.1.1. Elevated Immune Checkpoint Expression Correlates with Poor Patient Outcomes and Reduced Responses to Cancer Immunotherapy

Heightened expression and ligation of inhibitory receptors, such as programmed cell death protein 1 (PD-1), T cell immunoglobulin and ITIM domain (TIGIT), cytotoxic T-lymphocyte-associated protein 4 (CTLA-4), lymphocyte activation gene 3 (LAG-3), T cell immunoglobulin and mucin domain 3 (TIM-3), B and T lymphocyte attenuator (BTLA), CD244, and CD160, upon chronic antigen exposure in various solid cancers results in limited T cell survival and function [11,12,13,47,48,51,54,56,63,81,82,84,88,89,90,91,92,93,94,95,96,97,98,99]. Furthermore, when compared to peripheral blood lymphocytes, TILs of solid cancers demonstrate higher expression of inhibitory receptors and a more dysregulated phenotype [81,90]. Notably, expression of PD-1 on T cells correlates with higher co-expression of other inhibitory receptors and a more exhausted phenotype [47,63]. In HNSCC patients, high expression of TIM-3, an immune checkpoint receptor that activates cell proliferation via the Akt/S6 pathway, is a marker of exhausted TILs [48]. Interestingly, TIM-3 expression alone does not dictate TIL exhaustion [47]. While increased expression of TIM-3 is a marker of a more extensively exhausted phenotype, co-expression with PD-1 is required for greater TIL dysfunction, further emphasizing the essential role of PD-1 in T cell exhaustion [47]. High expression of PD-1 in both TIM-3^–^ and TIM-3^+^ TILs is associated with higher levels of B-lymphocyte-induced maturation protein-1 (BLIMP-1 and basic leucine zipper ATF-like transcription factor (BATF), transcription factors that impair T cell proliferation and cytokine secretion [47]. Co-expression of these receptors results in cross-talk that dampens Akt/S6 phosphorylation and regulates antitumor T cell responses [47,48].

It is evident that increased expression of inhibitory checkpoints contributes to T cell exhaustion of endogenous tumor-reactive T cells and therefore impacts the intrinsic antitumor capability of T cells. In addition, it has been found that elevated expression of negative regulators also facilitates exhaustion of adoptively transferred engineered T cells, consequently impairing the efficacy of adoptive cell therapy (ACT). Exhaustion is a major mechanism of tumor escape in relapsed cancers following CAR T cell treatment [85], and markers of T cell exhaustion are often associated with poor patient outcomes. In B cell acute lymphoblastic leukemia (B-ALL) patients treated with CD19-targeted, 4-1BB-based CAR T cells, significant DNA methylation reprogramming was observed in CD8+ CD19 CAR T cells four weeks post-infusion, notably methylation of genes associated with effector function and memory potential alongside demethylation of exhaustion-associated transcription factors BATF and thymocyte-selection-associated high mobility group box factor (TOX) [100]. In head and neck cancer, both clinical data and murine models demonstrate that the expression of PD-1 on TILs correlates with clinical outcomes and response to immunotherapy, with tumor rejection in response to anti-PD-1 monoclonal antibody (mAb) therapy associated with a decrease in PD-1^high^ and an increase in PD-1^low^ CD8+ TILs [101]. In chronic lymphocytic leukemia patients treated with CAR T cells, the enrichment and persistence of TIM3^+^PD-1^+^ CAR T cells were found to be associated with inferior clinical outcomes [102]. Even before infusion, CAR T cell products that have higher expression of inhibitory receptors, such as PD-1 and LAG-3, exhibit poorer in vivo expansion and antitumor activity in various leukemia and lymphoma patients [103,104].

Apart from cell-surface inhibitory receptors, altered intracellular negative regulators also play a role in T cell dysfunction in cancer. Decreased expression of Src-homology-2-domain-containing tyrosine phosphatase 1 (SHP-1), a cytosolic tyrosine phosphatase, results in hyperactive tyrosine kinase 2 (TYK2) and Janus kinase [105], kinases that facilitate lymphomagenesis [106,107]. In a murine melanoma model, a lower level of phosphorylated Src-homology-2-domain-containing tyrosine phosphatase 2 (SHP-2) was observed and correlated with increased metastasis, heightened release of inflammatory cytokines, and accumulation of myeloid-derived suppressor cells (MDSCs) in the TME [108].

#### 2.1.2. Changes in the Transcriptional and Epigenetic Landscapes of Dysfunctional T Cells and the Implications for T Cell Immunotherapy

Dysfunctional tumor-associated T cells display characteristic changes in their transcriptional landscapes. T cell exhaustion is epigenetically encoded, with exhausted T cells exhibiting reduced chromatin accessibility of T cell effector genes and increased accessibility of genes encoding exhaustion-related transcription factors and inhibitory receptors [85,109,110]. These epigenetic modifications result in various molecular cascades and transcriptional changes that exacerbate the dysfunctional state of TILs. For example, exhausted CD8+ TILs exhibit enriched chromatin accessibility of consensus binding motifs of nuclear factor of activated T cells (NFAT), a transcription factor that directly activates the anergy-inducing proteins gene related to anergy in lymphocytes (GRAIL) and Caspase-3 [47,110,111].

The members of the nuclear receptor subfamily 4A (NR4A1, NR4A2, and NR4A3), all of which are hormone-inducible transcription factors, are additional key regulators in the induction of T cell dysfunction and exhaustion [110,112,113,114,115,116]. Expression of NR4A1, the most extensively studied member of the NR4A family, is triggered by chronic antigen stimulation in solid tumors [115] and remains stable at high levels in tolerant T cells [113]. NR4A1 binds to and inhibits activator protein 1 (AP-1) and nuclear factor-κB (NFκB), thereby repressing IL-2 promoter activation and other cytotoxic effector gene expression [112,113]. NR4A1 further induces T cell dysfunction by promoting H3K27 acetylation and activating tolerance-related genes [113]. CD8+ T cells isolated from cancer patients exhibit high levels of NR4A transcription factors and enrichment of NR4A-binding motifs in accessible chromatin regions [95,109,114,117]. Expression of the NR4A family facilitates exhaustion of not only endogenous CD8+ TILs but also CD8+ CAR+ TILs. This highlights the potential of NR4A as a promising gene-editing target to improve T-cell-based therapies [110,115]. Indeed, studies performed by Chen et al. have demonstrated that triple knockout of NR4A1, NR4A2, and NR4A3 (Nr4a TKO) in CD19-targeted CAR T cells enhances tumor regression in and survival of tumor-bearing mice [115]. Nr4a TKO CAR+ TILs exhibited phenotypic, genotypic, and epigenetic profiles that were similar to those of functional effector CD8+ T cells, further highlighting the significant role that NR4A family members play in T-cell-intrinsic exhaustion [115].

Similar to NR4A1, TOX and TOX2 are key transcription factors highly expressed in TILs that program and maintain exhaustion in both endogenous and adoptively transferred CD8+ T cells [54,55,57,98,118,119,120,121,122]. TOX expression is induced by chronic TCR stimulation and downstream calcineurin and NFAT activation [118,119] and leads to T cell dysfunction by attenuating T cell effector functions and upregulating inhibitory receptors [119]. TOX is highly expressed in TILs, and its expression positively correlates with that of other exhaustion markers, such as PD-1 [118,119]. In HCC, TOX promotes the exhaustion of tumor-antigen-specific CD8+ T cells by inhibiting PD-1 degradation and therefore increasing PD-1 expression on TILs [55,57]. Significantly, TOX is critical for the induction of CD8+ T cell exhaustion; in the absence of TOX, exhausted T cells do not form [54,120]. This critical trait makes TOX another attractive gene-editing target in the development of T-cell-based immunotherapies, a concept actualized by Seo et al. in 2019 [122]. They demonstrated that, compared to wild-type CAR+ TILs, CAR+ TILs with the double knockout of TOX and TOX2 (Tox DKO) exhibited a higher level of effector cytokine production, a decrease in inhibitory receptor expression, and a superior capacity to suppress tumor growth and prolong the survival of tumor-bearing mice [122].

T-box expressed in T cells (T-bet) is a transcription factor that acts as a master regulator of polarization of Th1 T cells, a subset of CD4+ T cells with a high antitumor activity that positively correlates with improved prognosis in patients [123,124,125,126]. T-bet upregulates proinflammatory pathways, directly represses PD-1 expression, and is associated with robust and functional T cells [54,127,128,129]. Decreased levels of T-bet are associated with various malignancies [54,128,129]. In a study by Gacerez and Sentman, overexpression of T-bet in B7H6-targeted CD4+ CAR T cells increased proinflammatory and Th1 cytokine secretion and cellular cytotoxicity against B7H6-expressing tumor cells in vitro, and prolonged survival in an in vivo mice model [124]. These findings suggest that boosting T-bet activity may improve the antitumor efficacy of CAR T cell therapies against solid cancers.

Other exhaustion-related transcription factors, including BLIMP-1, BATF, interferon regulatory factor 4 (IRF4), and Eomesodermin (EOMES), have also been found to be overexpressed in exhausted CD8+ T cells [47,130,131,132]. Interestingly, both increased and decreased levels of BATF and IRF4 in TILs have been linked to oncogenesis [132,133]. Overexpression of BATF and IRF4 has been observed in exhausted T cells and functions to promote exhaustion and repress memory T cell differentiation [132]. By contrast, Seo et al. demonstrated in a murine model that overexpression of BATF and IRF4 in CAR T cells promoted T cell survival, proliferation, memory differentiation, and production of effector cytokines [133]. These contrasting findings may be due to the different models used in the two studies, the first of which analyzed T cell exhaustion in the context of chronic infection, whereas the second focused on the exhaustion of CAR T cells in a tumor mouse model.

T cell signaling and functionality are also impacted by the expression of various phosphatases that act in TCR-initiated signaling pathways. For example, dual specificity phosphatase 2 (DUSP2), a mitogen-activated protein kinase (MAPK) phosphatase that regulates phospho-signaling downstream of the TCR, is highly expressed in tumor-infiltrating CTLs and negatively correlates with cancer patient survival [134]. In terms of its mechanism, DUSP2 recruits the Mi-2β nucleosome-remodeling and histone deacetylase complex, resulting in chromatin remodeling that facilitates T cell exhaustion and dysfunction [135].

Another phosphatase, protein tyrosine phosphatase N2 (PTPN2), has also been linked to T cell dysfunction. PTPN2 contributes to the generation of terminally exhausted CD8+ T cells and negatively impacts αβ TCR signaling. It interferes with cytokine signaling that is essential for normal T cell function, homeostasis, and differentiation by dephosphorylating and inactivating target proteins such as LCK, JAK-1, and JAK-3 from the Janus-activated kinase [105] family, and STAT-1, STAT-3, and STAT-5 from the signal transducer and activator of transcription [136] family [137,138,139,140,141,142]. In a study conducted by Wiede et al. using a mice model, the deletion of PTPN2 in HER-2-specific CAR T cells led to an increase in LCK activation and cytokine-induced STAT-5 signaling, resulting in enhanced CAR T cell activation, homing to, and eradication of target HER-2+ mammary tumors in vivo [137].

Notably, various transcription and epigenetic factors could also play a cooperative role in modulating the dysfunctional T cell phenotype. Through single-cell sequencing of TILs isolated from HCC patients, a variety of transcription factors (e.g., EOMES, T-bet, BLIMP-1, v-maf avian musculoaponeurotic fibrosarcoma oncogene homolog (MAF), TOX, and T cell factor 7 (TCF7)) were shown to regulate CD8+ T cell dysfunction in a complex, interdependent manner [54]. The presence of certain transcription factors, including recombination signal binding protein for immunoglobulin kappa J region (RBPJ), zinc finger BED-domain-containing protein 2 (ZBED2), ETS variant transcription factor 1 (ETV1), inhibitor of DNA binding 3 (ID3), MAF, BLIMP-1, and EOMES, was also determined to be a characteristic of dysfunctional CD8+ T cells in a study conducted with melanoma patients [82]. In lymphomas, the epigenetic modifiers ten-eleven translocation 2 (TET2), DNA methyltransferase 3A (DNMT3A), and isocitrate dehydrogenase 2 (IDH2) are commonly mutated and correlate with heightened hypermethylation of genes involved in TCR signaling and T cell differentiation [75,77,78,143,144,145]. The key altered negative regulators and transcription and epigenetic factors in T cells that contribute to T cell exhaustion and dysfunction are summarized in Table 2.

### 2.2. Cancer-Cell-Intrinsic Characteristics Hinder Cancer Immunosurveillance and Limit the Efficacy of T-Cell-Based Cancer Immunotherapies

In addition to endogenous T cell dysfunction caused by negative regulator upregulation and changes in the transcriptional and epigenetic landscapes, cancer-cell-intrinsic characteristics also contribute to immune dysfunction and cancer progression (Figure 1).

#### 2.2.1. Molecular Alterations in Tumor Cells

In cancer cells, molecular alterations in key signaling pathways, proto-oncogenes, tumor suppressors, and epigenetics all contribute to cancer development and progression (Table 3).

##### Key Signaling Pathways Altered in Tumor Cells

Alterations in a myriad of intracellular signaling pathways can contribute to cancer development. In a range of cancer types, genetic alterations in several key signaling pathways have been identified, including the Ras-Erk and PI3K-Akt signaling pathways [222,223,224]. Studies using next-generation sequencing technology conducted with thousands of tissue samples and computational scanning of comprehensive cellular pathway databases have revealed the presence of alterations in additional pathways, such as the Nrf2, Wnt, Myc, p53, Hippo, and Notch pathways [225,226,227,228].

##### Alterations of Proto-Oncogenes and Tumor Suppressors

Under normal physiological conditions, proto-oncogenes and tumor suppressors act in opposing ways to facilitate homeostatic cell growth [229,230]. Proto-oncogenes stimulate cell division and promote normal cellular growth [230]. Activation via gain-of-function converts proto-oncogenes into oncogenes, which are commonly found in cancer, and results in uncontrolled cell proliferation [230]. Examples of commonly mutated proto-oncogenes include RAS (KRAS, NRAS, and HRAS) and BRAF in the RNA/RAF/MEK/MAPK signaling pathway [230]. Conversion of proto-oncogenes to oncogenes can result from single mutations, such as the common G12V mutation of KRAS, as well as gene amplifications, as seen with c-MYC [231,232,233,234,235,236].

By contrast, tumor suppressors inhibit cell division and prevent cell growth [229]. Inactivation of tumor suppressors via loss-of-function mutations results in a loss of control of cell division and evasion of apoptosis, which contribute to tumorigenesis [229]. A typical and well-studied tumor suppressor is tumor protein p53 (TP53), a transcription factor known as the “guardian of the genome” [237]. TP53 tightly controls cell division and apoptosis, and, in response to DNA damage during cell division, promotes DNA damage repair or apoptotic pathways [237]. Mutations in TP53 can lead to the loss of its tumor suppressor function or the gain of oncogene function [238]. Germline mutations in TP53 cause Li-Fraumeni syndrome [239], a condition in which patients develop early onset breast cancer, sarcomas, or other neoplasms [240].

##### Common Epigenetic Alterations Identified in Cancer Cells

Epigenetic changes inextricably cooperate with genetic alterations to drive the malignant cancer phenotype. Several factors contribute to epigenetic changes, including DNA methylases/demethylases, histone-modifying enzymes, histone posttranslational modification readers, chromatin/nucleosome remodelers, and microRNA (miRNA) [241,242]. Hypermethylation in the promoter region of tumor suppressors is commonly seen in cancers. For example, hypermethylation of retinoblastoma protein (RB), phosphatase and tensin homolog (PTEN), breast cancer type 1 susceptibility protein (BRCA1), breast cancer type 2 susceptibility protein (BRCA2), O6-methylguanine-DNA methyltransferase (MGMT), cyclin-dependent kinase inhibitor 2B (CDKN2B), and Ras-association-domain-containing protein 1A (RASSF1A) are all reported in cancer, and result in the loss of ability to protect against tumorigenesis [243,244,245,246]. Moreover, histone modifications mediated by histone acetyltransferases (HATs), deacetylases (HDACs), methyltransferases (HMTs), and demethylases (HDMs) at regulatory regions can also lead to aberrant gene expression via the introduction or removal of acetyl or methyl groups, respectively [247,248,249]. For instance, the removal of acetyl groups from the promoter of the CDKN1A gene by HDACs leads to its downregulation and is linked to the onset of various cancers [202]. Finally, alterations in chromatin remodeling complexes are also responsible for the development of cancer. Changes in the switch/sucrose non-fermenting (SWI/SNF) complex are often observed in melanomas and renal carcinomas and correlate with poor survival rates [250]. Mutations in the AT-rich interactive-domain-containing protein 1A (ARID1A) gene, which encodes BAF250a, a component of the SWI/SNF family of chromatin remodeling complexes, are frequently observed in ovarian clear cell carcinomas, endometrioid carcinomas, pancreatic adenocarcinomas, transitional cell carcinomas, and triple-negative breast cancers [208,209,251].

#### 2.2.2. Immunosuppressive Tumor Microenvironment

##### Metabolic Competition in the TME

Similarities can be observed between the metabolism of cancer cells and activated T cells due to their shared characteristics of rapid growth and proliferation [252]. While most cell types prioritize oxidative phosphorylation as the primary energy production pathway in the presence of oxygen, both cancer cells and activated lymphocytes exhibit “aerobic glycolysis” as their main metabolic program, whereby they preferentially utilize glycolysis for energy production even when sufficient oxygen is available to support oxidative phosphorylation [252,253]. This preference for glycolysis over oxidative phosphorylation is driven by the need for rapid biosynthesis, which glycolysis can efficiently achieve, whereas oxidative phosphorylation emphasizes efficient ATP production [252]. This similarity in metabolic profiles between cancer cells and T cells is the foundation of metabolic competition in the TME [254]. The metabolic restriction of T cells caused by tumor glucose consumption leads to decreased mTOR activity, glycolytic capacity, and IFN-γ production in T cells, ultimately facilitating T cell dysfunction and enabling tumor progression [254].

##### Nutrients and Metabolites

Cancer cells characteristically exhibit an elevated glycolysis rate, which consequently depletes the TME of the glucose required for normal T cell effector function and hence impairs the immune response against cancer [254,255]. In some cancers, such as squamous cell carcinoma and pancreatic adenocarcinoma, this increase in glycolysis is facilitated by the overexpression of glucose transporter 1 (GLUT-1) [256,257]. The high glucose uptake and glycolysis rate exhibited by cancer cells correlate with lower CD8+ T cell infiltration and poorer patient survival [256,257]. In terms of the effect on T cells, a lack of glucose hinders T cell effector functions by inhibiting mammalian target of rapamycin complex 1 (mTORC1) activation and downstream Myc induction, impairing NFAT signaling, and suppressing IFNγ translation [258,259]. Significantly, glucose depletion does not impair the differentiation or function of Tregs; in fact, glucose-deficient environments promote Treg induction and activity in vitro [260]. Collectively, these observations demonstrate that tumor-cell-mediated glucose depletion enhances the immunosuppressive character of the TME by categorically favoring Treg survival and function [255,260].

Cells also compete for glutamine in the TME, an essential nutrient for tricarboxylic acid (TCA) cycle maintenance and lipid synthesis in both cancer cells and effector T cells [261,262,263]. In various cancers, alanine, serine, and cysteine transporter 2 (ASCT2), the major glutamine transporter, are overexpressed, giving these cancer cells an advantage over other cells in glutamine uptake [264].

Arginine is an essential nutrient for the urea cycle and polyamine synthesis and is required by both cancer and T cells [261]. A variety of cancer cells overexpress nitric oxidase synthase and arginase, major enzymes in arginine metabolism, therefore allowing them to outcompete other cell types for arginine uptake [265,266]. Evidence from mouse models suggests that increasing the L-arginine concentration can result in a shift in T cell metabolism from glycolysis to oxidative phosphorylation, promoting the generation of central-memory-like T cells with enhanced antitumor activities and longer persistence [267].

Indoleamine-2,3-dioxygenase (IDO), an enzyme that converts tryptophan to kynurenine, has elevated activity in various cancers, resulting in a depletion of the essential nutrient tryptophan and an accumulation of by-product kynurenine in the TME [268,269,270]. Tryptophan deficiency impairs T cell effector function by inhibiting mTORC1 activity and downregulating TCR-CD3-ζ [271]. Furthermore, elevated IDO activity increases the level of kynurenine in the TME, and this metabolite impairs TCR signaling and therefore facilitates T cell dysfunction [255].

Tumor growth can lead to a decrease in oxygen perfusion and result in a hypoxic TME, a signature of advanced cancer [255]. Hypoxic environments promote tumor progression by facilitating CD8+ T cell dysfunction via various mechanisms [272]. Hypoxia directly impacts T cells by inducing them to differentiate into FoxP3^+^ Treg cells, which, under hypoxic conditions, produce extracellular adenosine that further suppresses T cell effector function [272,273,274,275]. Additionally, exposure to hypoxia in the TME increases histone bivalency in CD8+ TILs, leading to a reduction in their transcriptional potential and contributing to the development of T cell exhaustion [276]. Hypoxia also directly contributes to T cell exhaustion by driving T cell expression of CD39, an ectoenzyme that secretes immunosuppressive adenosine [277]. Furthermore, hypoxia also indirectly hinders antitumor immunity by impacting tumor-associated immune and stromal cells. Under hypoxic conditions, tumor cells and associated cells secrete cytokines and factors that attract Tregs to the TME [278]. Hypoxic conditions within the TME also stimulate the expression of hypoxia-inducible factors (HIFs) in tumor cells [272,279,280]. HIF-1α induces the expression of vasculature endothelial growth factor (VEGF), a pro-angiogenic factor that promotes the formation of abnormal tumor vasculature characterized by a deficiency in pericytes and a leaky basement membrane [272,279,280]. Consequently, this prevents lymphocyte extravasation from blood vessels into the TME and facilitates immune exclusion [272,279,280]. HIF-1α also stimulates the expression of the immune checkpoint inhibitors PD-L1, CTLA-4, and LAG-3 on tumor and tumor-associated immune and stromal cells [272,279,280]. In the TME, low oxygen levels also inhibit T cell proliferation by affecting potassium concentration, although the exact mechanism is unknown. Hypoxia has been shown to reduce the expression of Kv1.3 potassium channels in T cells, which play a crucial role in maintaining the membrane potential and function of T cells [281]. However, another hypothesis suggests that necrotic tissue in the TME contributes to high potassium concentration in the extracellular fluid, resulting in AKT/mTOR hypophosphorylation downstream of the TCR and significantly suppressing T cell effector function [282]. This presents a paradoxical contrast to the first hypothesis. Baixauli and Vodnala found that a high K^+^ concentration can trigger stem-cell-like characteristics in TILs, rendering them more effective in mediating tumor destruction following immune checkpoint inhibitor therapy [283,284]. Accordingly, highly proliferative TILs, characterized by elevated expression of the proliferation marker Ki67, also express high levels of Kv1.3 channels [285].

Calcium (Ca^2+^) is also critical to various cellular functions, including signaling, metabolism, apoptosis, and cell division [286,287,288]. Increased calcium levels in cancer cells confer a growth advantage, impairing calcium uptake by T cells in the TME and consequently hindering T cell activation and proliferation [289,290,291]. Using targeted optogenetic stimulation of intracellular Ca^2+^ signaling, Kim et al. successfully restored the cytotoxic effector functions of T cells in tumor sites [292].

##### TME-Derived Metabolites

Reactive oxygen species (ROS) are mainly generated as a by-product of energy production and are known to be markers of cell stress and inducers of cellular damage and death. ROS also affect T cell metabolism and are therefore key modulators of the antitumor activity of T cells. Sustained high levels of ROS are linked to immunosuppression in the TME [293,294]. They can impede T cell function in a multitude of ways, including inhibition of the DNA-binding capacities of NFAT and NKκB [295] and disruption of pMHC–TCR binding [296]. In the TME, various tumor-associated immune and stromal cells contribute to high levels of ROS. For example, immature myeloid cells, myeloid-derived suppressor cells, and activated granulocytes produce ROS, which results in apoptosis and suppression of T cells [294,296,297,298]. Interestingly, while high levels of ROS are detrimental to T cells, moderate levels of ROS are required for normal TCR signaling and T cell activation [293,294,295].

Elevated levels of cholesterol within tumors also contribute to the immunosuppressive environment of the TME. High cholesterol increases endoplasmic reticulum (ER) stress in cytotoxic T cells and thereby induces upregulation of inhibitory receptors in T cells, including PD-1, 2B4, TIM-3, and LAG-3 [299].

Additionally, excessive secretion and subsequent accumulation of lactic acid by tumor cells lowers the pH of the TME. The acidic environment negatively affects T cell effector function by suppressing the PI3K/Akt/mTORC1 pathway and glycolysis, pathways that are critical for normal T cell activation, proliferation, and function [300,301,302]. However, Tregs have a metabolic advantage over conventional T cells due to the action of the Treg transcription factor FoxP3, which reprograms Treg metabolism to better adapt to the lactic-acid-rich TME. FoxP3 suppresses Myc and glycolysis, enhances oxidative phosphorylation, and increases nicotinamide adenine dinucleotide (NAD) oxidation, optimizing the metabolic profile of Tregs to suit the low-glucose, high-lactate TME [303].

##### Tumor-Associated Immune and Stromal Cells

Various resident or recruited immune and stromal cells within the TME significantly contribute to the immunosuppressive environment. The dual influence of these cells on both cancer progression and effector T cell dysfunction is illustrated in Figure 2.
Tumor-Associated Macrophages (TAMs)

Tumor-associated macrophages (TAMs), infiltrating macrophages characterized by a pro-tumorigenic M2-like phenotype, are present in solid tumors and critically contribute to the pro-tumoral environment of the TME by facilitating tumor growth, angiogenesis, and metastasis while attenuating antitumor immune responses [304,305,306,307]. TAMs accumulate and are retained in the hypoxic TME due to the presence of macrophage chemoattractants (e.g., monocyte chemoattractant protein (MCP)-1 and macrophage inflammatory protein (MIP)-1α) [308,309]. TAMs advance tumor initiation, progression, and metastasis by secreting immunosuppressive cytokines, chemokines, proteolytic enzymes, and growth factors and by upregulating immune checkpoints on T cells [307]. IL-6, IL-23, and IL-17 secreted by TAMs promote inflammation within the TME that drives tumor growth [310,311]. TAMs further facilitate metastasis by promoting epithelial–mesenchymal transition (EMT) via the secretion of IL-1β, IL-8, TNF-α, and TGF-β [312,313,314]. Matrix metalloproteinases (MMPs), cathepsins, and serine proteases produced by TAMs mediate tumor-cell–ECM interactions and ECM degradation, thereby aiding the intravasation of tumor cells [315,316,317]. TAMs also play a role in tumor angiogenesis and the remodeling of existing vasculature via the production of factors such as MMP-9, VEGF, FGF-2, CXCL8, IL-1, IL-18, COX-2, iNOS, and MMP7 [318,319,320,321,322]. In addition to facilitating the progression and metastasis of cancer cells within the TME, TAMs also promote the survival of circulating tumor cells (CTCs) by activating the PI3K/Akt survival signaling pathway in cancer cells and secreting protective chemokines and cytokines [323,324,325]. In clinical studies, a high TAM count has been found to correlate with a poorer prognosis in patients with various cancer types [136,326,327,328].

TAMs facilitate CD8+ T cell dysfunction via several mechanisms. TAM-derived secretory factors, including ARG1, iNOS, TGF-β, IL-10, and ROS, induce CD8+ T cell exhaustion and dysfunction [329,330]. TGF-β elevates the expression of the checkpoint inhibitory molecules TIM-3, PD-1, and CTLA-3 on T cells and inhibits IFNγ and granzyme B secretion [331]. Furthermore, TAMs express PD-L1 and induce its expression on monocytes, which promotes T cell exhaustion upon ligation [330,332]. TAMs further attenuate effector T cell function by secreting high levels of IDO and depleting the TME of essential amino acids, including L-arginine and tryptophan [332].
Cancer-Associated Fibroblasts (CAFs)

Various signaling molecules within the TME, including TGF-β, ROS, inflammatory cytokines, contact signals, and receptor tyrosine kinase (RTK) ligands, trigger the conversion of fibroblasts into activated cancer-associated fibroblasts (CAFs) that exhibit significant heterogeneity and plasticity [333,334]. CAFs affect the production and structure of the ECM and secrete factors that promote tumor growth and suppress antitumoral immunity. CAFs deposit and remodel the ECM by secreting matrix-crosslinking enzymes, which contribute to the structural stiffness of tumor tissue [333,335,336,337]. The rigidity of tumor tissue promotes cancer growth by triggering survival and proliferation signaling within cancer cells and impedes immune cell infiltration and function by collapsing vasculature and creating a hypoxic environment [337,338,339,340]. Interestingly, CAFs also induce angiogenesis in a manner that selectively promotes tumor cell growth [341]. CAF-derived matrix proteases mediate tumor matrix remodeling that excludes CD8+ T cell infiltration, promotes local cancer cell invasion, and increases the metastatic potential of cancer cells [341,342,343,344,345]. Several matrix components, such as tenascin and periostin, also enhance cancer cell survival by increasing Wnt signaling [346,347]. Tumor growth and survival are further supported by CAF-derived growth factors, cytokines, and exosomes, including TGF-β, LIF, GAS6, FGF5, GDF15, and HGF, which collectively act to promote the invasive and proliferative potential of cancer cells [348,349,350,351,352,353,354].

CAFs indirectly facilitate T cell dysfunction by secreting CXCL1 and CXCL2, which polarize macrophage differentiation into generating the immunosuppressive M2 and TAM subtypes [332]. Ligation of the CAF inhibitory ligands PD-L2 and FasL to their corresponding receptors on CD8+ T cells directly induces effector T cell dysfunction [332]. The cytokines and chemokines secreted by CAFs also influence a range of immune cells, resulting in both immunosuppressive and immunoenhancing outcomes [341]. Although CAFs play a dual role in immune response regulation, they predominantly create an immunosuppressive environment. For example, IL-6, CXCL9, and TGF-β secreted by CAFs attenuate T cell effector responses [333,355]. CAFs have also been shown to engage in antigen cross-presentation that activates CD4+ T cells and suppresses CD8+ T cells, an observation further supported by clinical studies [356,357,358]. This effector T cell suppression is mediated by PD-L1 and PD-L2 expression on CAFs, which inhibits the production of IL-2, a major T cell growth factor [359].
Tumor-Associated Neutrophils (TANs)

Tumor-associated neutrophils (TANs) are another class of cells closely associated with the TME. Like macrophages, TANs can polarize toward an anti-tumoral subtype (N1 TANs) or a pro-tumoral subtype (N2 TANs) [360,361,362,363,364]. N2 polarization is stimulated by the presence of TGF-β in the TME [361,365]. While TANs have been shown to exhibit both pro-tumoral and anti-tumoral functions, this section focuses on their pro-tumoral functions, including the promotion of angiogenesis, EMT transition, migration and invasion, and immunosuppression [366]. It is important to note that regarding the functions of TANs, contradictory evidence currently exists in the literature due to the high heterogeneity of the neutrophil population within the TME and insufficient markers being available to distinguish different subtypes [366]. Furthermore, a reliance on murine models has resulted in conclusions being drawn that are not necessarily relevant in clinical settings [367].

TANs facilitate tumor initiation via the actions of neutrophil elastase (NE), ROS, and reactive nitrogen species (RNS) [294,368,369,370]. They release and/or induce the production of genotoxic substances (e.g., ROS) that damage the DNA of epithelial cells and facilitate carcinogenesis [371]. Various proteases secreted by TANs, including NE and MMP9, activate growth factor pathways within cancer cells and enhance hyperproliferation [371,372]. TANs also generate neutrophil extracellular traps (NETs) composed of DNA–histone complex, proteins, and fibers secreted by activated neutrophils, and these facilitate tumor progression and metastasis by sequestering and promoting the adhesion of circulating cancer cells in distant organs [373,374,375]. TANs also play a protective role in the survival of CTCs. CTCs associated with neutrophils have been found to demonstrate greater proliferation and enrichment of positive regulators of cell cycle progression compared to CTCs alone [376]. Furthermore, TAN-derived MMP9 and VEGF facilitate angiogenesis [377,378]. TANs also trigger EMT via the NE cleavage of E-cadherin, upregulation of EMT-related genes (including TWIST and Vimentin), facilitation of nuclear translocation of β-catenin, and promotion of nuclear expression of ZEB1, a transcription factor that boosts tumor invasion and metastasis [379,380,381].

TAN-derived cytokines and chemokines also interact with other immune cells within the TME to promote an immunosuppressive environment. TANs suppress T-cell-mediated immunity by releasing high levels of arginase and protumor factors, including CCL2, CCL5, NE, and cathepsin G [377,382]. Additionally, TAN-derived CCL17 recruits Tregs to the TME, while CXCL14 attracts M2 macrophages [383]. ROS, H_2_O_2_, and ARG-1 produced by TANs further contribute to immunosuppression [361,384].

## 3. Exploring the Therapeutic Landscape for Cancer Treatment

Many traditional cancer treatments, such as chemotherapy and radiotherapy, have severe side effects and fail to specifically target cancer cells. Since the accomplishment of the human genome project, progress in the development of high-throughput, low-cost technologies has enabled significant advancement in the generation of novel cancer treatments, better prediction of patient outcomes, and the development of targeted therapies. A summary of the traditional therapies and the novel targeted approaches used in cancer treatment is shown in Figure 3.

### 3.1. Targeted Therapies: Small-Molecule Drugs, Therapeutic Antibodies, Oncolytic Viruses, and Gene Editing

Targeted therapies are a newer approach to cancer treatment designed to selectively destroy cancer cells, sparing normal cells and reducing toxic side effects [385]. More than 80 anticancer small-molecule drugs have been approved by the US FDA [386]. For example, various small-molecule drugs have been developed that successfully target the G12C mutation commonly seen in KRAS, a mutation previously deemed impossible to target due to its structural and biochemical properties [387,388,389,390].

Since the development of hybridoma technology in 1975 by Milstein and Köhler, therapeutic antibodies based on monoclonal antibodies (mAbs) or polyclonal antibodies (pAbs) have evolved and undergone a revolution [391]. Unlike small-molecule drugs that penetrate the cell membrane to block signaling, antibodies mainly bind to receptors or ligands on the cell surface and have various therapeutic mechanisms of action. For instance, antibodies can block the binding of signaling molecules, thereby inhibiting cell proliferation or facilitating cell death through the inhibition of DNA repair, prevention of cell cycle progression, and/or induction of apoptosis [392,393,394,395]. Therapeutic antibodies can also recruit immune effector cells to trigger antibody-dependent cellular cytotoxicity (ADCC), antibody-dependent cellular phagocytosis (ADCP), and complement-dependent cytotoxicity (CDC) [396].

Another promising new therapy is oncolytic virus therapy. Due to the lack of functional antiviral mechanisms in cancer cells, oncolytic viruses can be utilized to selectively replicate in and lyse cancer cells while sparing normal tissue [397,398]. The concept was initiated in the early 1900s, and in 2015, T-VEC became the first FDA-approved vaccine for the treatment of melanoma [399].

Gene-editing technologies are also being utilized in targeted therapeutic approaches to correct mutations in both cancer cells and T cells, thereby strengthening the antitumor response. The clustered regularly interspaced palindromic repeats (CRISPR)/Cas9 system is a widely used gene-editing tool known for its simplicity, broad applicability, and high potential [400]. In recognition of its impact, the CRISPR/Cas9 system was awarded the Nobel Prize in Chemistry in 2020 [400]. The CRISPR/Cas9 system is rapidly gaining popularity in the field of cancer immunotherapy. For instance, it can be utilized to correct mutations in the TP53 gene, thereby restoring its tumor-suppressing properties and reducing the growth of tumors [401]. Additionally, the use of multiple sgRNAs to target the human estrogen receptor 2 (HER2) gene has been shown to effectively decrease HER2 expression and impede the growth of HER2-positive cancer cells [402]. In addition to its application in editing sites within tumor cells, CRISPR/Cas9 technology has been widely utilized to restore and enhance T cell function. For example, CAR T cells that have been engineered to lack PD-1 or CTLA-4 exhibit increased cytotoxicity, robust cytokine release, and a potent antitumor response [403,404]. By targeting the integration of the CAR construct into the TCR α constant region, universal expression of CAR and prolonged T cell persistence in vivo can be achieved [405,406]. The technology can also be used to generate universal “off-the-shelf” CAR T cells by eliminating endogenous TCR, MHC, and CD52 molecules [407]. Furthermore, mutations in T cell signaling can be corrected to restore normal T cell function [408].

### 3.2. Significant Emerging Immunotherapies: Cancer Vaccines, Engineered T Cells, and Immune Checkpoint Inhibitors

Of the various advancements in cancer treatment over the past decade, immunotherapy stands out as a particularly promising approach. Its remarkable efficacy and specificity earned it the title of Science Breakthrough of the Year in 2013 [409]. Immunotherapy is a targeted cancer treatment that utilizes a patient’s own immune system to eliminate cancer cells [409]. Its effectiveness lies in its ability to target neoantigens, unique antigens generated by tumor-specific mutations [410]. By doing so, the immune system can differentiate between cancer and non-cancer cells, allowing it to specifically attack and destroy the former while leaving healthy tissue unharmed [410]. Several categories of cancer immunotherapies exist, including cancer vaccines, CAR T cells, TCR T cells, and TILs, all of which have demonstrated durable clinical responses.

In the past decade, several anti-neoantigen cancer vaccines have been developed [411]. The first cancer vaccine consisted of dendritic cells loaded with neoantigens and was successful in eliciting specific T cell responses in melanoma patients [412]. Sahin et al. then proved the feasibility of “individualized mutanome vaccines”—vaccines tailored to the mutations in each individual cancer patient—paving the way for personalized cancer immunotherapy [413]. Vaccines containing synthetic long peptides designed using bioinformatics have also been shown to improve the overall survival of cancer patients [414,415]. mRNA vaccines, known for their versatility in design and target selection, have also been widely used in cancer treatment and prevention [416]. For example, Rosenberg and colleagues developed an mRNA vaccine for gastrointestinal cancer patients that elicited neoantigen-specific T cell immunity [417]. This was demonstrated by the successful isolation and verification of T cells targeting KRAS G12D from treated patients, demonstrating the safety and efficacy of mRNA vaccines [417].

Furthermore, the incorporation of heat shock protein (HSP)-based components into cancer vaccines has been shown to improve treatment efficacy [418]. Heat shock proteins (HSPs), whose expression is induced by heat shock or other cellular stressors, are a large family of proteins involved in protein folding and trafficking that also play a pathological role in cellular proliferation, differentiation, and carcinogenesis [418]. Because HSPs are significantly overexpressed in a wide range of human cancers and have the promiscuous ability to chaperone and present a broad repertoire of tumor antigens to APCs, they are used in cancer immunotherapy as a means of promoting cancer antigen-specific T cell stimulation [419,420,421]. When cancer cells undergo stress, such as during chemotherapy or radiation therapy, they release HSPs into the extracellular space [422]. These extracellular HSPs, in complex with tumor-associated antigens, can then bind to receptors expressed on APCs, leading to endocytosis, antigen cross-presentation to MHC I molecules, T cell priming, and induction of antigen-specific CTLs [422,423,424]. Cancer vaccines incorporating HSP–peptide complexes have demonstrated successful elicitation of immune responses directed specifically against the tumor from which the HSPs were isolated [422,425,426,427].

The development of therapeutic antibodies has greatly contributed to the advancement of various immunotherapies. Antibodies that mimic TCR function have been successfully developed due to the high efficiency with which peptide–MHC complexes can be purified and refolded in vitro [428,429,430,431,432,433]. The versatility of antibodies has enabled their modification for different purposes, expanding their use in cancer immunotherapy, such as in the design of bi-specific T cell engagers (BiTE), antibody–drug conjugates (ADC), and CAR T cells [431,434,435,436,437,438,439,440]. Immune checkpoints, such as CTLA-4 and PD-1, play a role in negatively regulating the immune response. Antibody therapies that target these checkpoints and their ligands have been shown to produce durable and persistent clinical responses in various cancer patients through the restoration of T cell function [441].

### 3.3. Challenges and Emerging Strategies in Cancer Immunotherapy

#### 3.3.1. Combating Antigen Escape in CAR T Cell Therapy

Despite the progress made in cancer immunotherapy, various obstacles still need to be addressed. For example, antigen escape due to the loss or mutation of the target antigen is commonly seen in relapsed patients treated with CD19-targeting CAR T cell therapy [442]. Identifying target antigens that are consistently and exclusively expressed above a detectable threshold on tumor cells, while not being expressed on essential healthy cells, is a critical challenge in developing safe and efficacious CAR T cell therapies for solid tumors [443]. To combat the challenge posed by antigen escape, various pharmaceutical strategies have been developed that can selectively upregulate the expression of cell-surface target antigens on cancer cells, thereby increasing target antigen density recognition by CARs [443]. For example, the use of epigenetic modulators as sensitizers for CAR T cell therapy has been successfully demonstrated in multiple preclinical studies. Anurathapan and colleagues established that exposure to decitabine, a DNA methyltransferase inhibitor, upregulates cell-surface expression of mucin short variant S1 (MUC1) on pancreatic cancer cells, thus making tumor cells more susceptible to in vitro cytolysis by MUC1-specific CAR T cells [444]. Besides epigenetic modulators, other compounds, including protein kinase C modulators [445] and γ-secretase inhibitors [446], have been demonstrated to successfully upregulate antigen expression on target cells and abate immune escape of tumor cells expressing low levels of the target antigen.

##### Multiple Antigen-Targeting Strategies in CAR T Design

While the upregulation of a single target antigen can increase the efficacy of CAR T cell therapy against certain malignancies, the highly heterogenous nature of most cancers renders single-antigen-targeting strategies suboptimal in many cases. To tackle the challenge of phenotypic heterogeneity, various innovative CAR T therapies designed to target multiple antigens are being developed.

##### Combination of Single-Antigen Specific CAR T Therapies

The most straightforward method entails the co-administration of two single, specific CAR T cell products, such as both CD19-targeted and CD22-targeted CAR T cells, either sequentially [447] or simultaneously [448]. Bispecific CAR T cells have also been developed, which coexpress two different CAR transgenes and are activated by recognition of either antigen A or antigen B [449].

##### Tandem CARs

Another design that allows multiple antigen targeting involves tandem CAR T cells, which combine the antigen-recognition scFvs for two distinct antigens in a manner so that the combined exodomain can co-engage both antigens together in a bivalent immune synapse [450,451,452]. Tandem CARs exhibit T cell activation upon recognition of either antigen, and binding to both targets results in an even more potent T cell response [443].

##### Combination of CAR T Cells and BiTEs

In another innovative method, CAR T cell targeting is combined with the release of bispecific T cell engagers (BiTEs), fusion proteins that consist of two scFvs—one that binds to T cells via the CD3 receptor, and the other that recognizes a tumor-specific antigen [453]. Upon activation, these CAR T cells synthesize and release BiTEs, which subsequently recruit bystander T cells to elicit an immune response against a second tumor-associated surface antigen [454].

##### Modular CAR T Designs

Finally, the invention of modular CAR T designs has greatly increased the flexibility and ease of design, enabling the generation of a large panel of CARs against numerous tumor-associated antigens [455,456,457]. In this approach, T cells are genetically modified to express a ‘universal’ CAR that targets a non-human molecule, such as biotin or fluorescein isothiocyanate (FITC), instead of a tumor-associated antigen [443]. Subsequently, these CAR T cells can be activated by administering bispecific adapters, such as an FITC-labeled tumor antigen, that cross-link the CAR T cells to the specified antigen [443]. The modular design offers a significant benefit by enabling the sequential or simultaneous infusion of various adapters to target a broad range of heterogeneously expressed antigens [443,456].

#### 3.3.2. Addressing Additional Challenges: T Cell Trafficking and Persistence

Another challenge to overcome is poor T cell trafficking, which contributes to the lack of success of CAR T cells against solid tumors, despite its remarkable response rates against hematological malignancies. To ameliorate T cell trafficking to and penetration of the tumor site, one straightforward solution is to perform local administration of T cells directly into the tumor site, rather than systemically via intravenous injection [458]. Another approach used to improve T cell trafficking involves engineering T cells to express chemokine receptors that match the chemokines present in the TME [459]. For example, CAR T cells engineered to overexpress CXCR1 or CXCR2 demonstrated significantly enhanced T cell trafficking and antitumor responses [460,461]. Additional chemokine receptors, including CCR2, CCR5, CCR6, and CXCR3, are known to be used for effector T cell recruitment to the tumor site, indicating their potential for use in therapeutic applications [462,463].

A lack of long-term persistence following the adoptive transfer of CAR T therapies presents another significant barrier in the development of effective CAR T therapies against solid cancers. A plethora of factors have been shown to contribute to the poor long-term persistence of adoptively transferred T cells, including patient preconditioning, ex vivo culture conditions, development of T cell exhaustion, design of the CAR construct, and host immune response against the infused product [464,465,466,467,468,469]. In order to achieve long-term persistence, adoptively transferred T cells must infiltrate secondary lymphoid organs to continually renew the population of circulating T cells that subsequently migrate to the tumor sites [470,471]. A method to improve persistence was elucidated by Xu and colleagues, who successfully demonstrated that the addition of IL-7 and IL-15 in cell culture media during ex vivo expansion of CAR T cells increased the frequency of CD8+ T cells positive for CD45RA, a marker of naïve T cells, and CCR7+, a chemokine receptor essential for T cell homing to secondary lymphoid organs [470]. This subset of CD8+CD45RA+CCR7+ T cells exhibited robust antitumor activity even with repetitive antigen challenges and maintained migration to secondary lymphoid organs, therefore achieving prolonged persistence [470].

Furthermore, solid cancers present the additional challenge of tumor-type-specific mechanisms that hinder the therapeutic efficiencies of CAR T therapies. Different solid cancers have distinct characteristics, such as an exceptionally dense tumor stroma, heterogenous expression of cancer antigens, or lack of strong immunogenic neoantigens—factors that deter the ability of CAR T cells to infiltrate the tumor and recognize and eliminate cancer cells [472,473]. For example, pancreatic ductal adenocarcinoma (PDAC) is characterized by an extremely dense stroma deposition surrounding legions that presents a physical barrier against immune infiltration [474]. In glioblastoma, profound tumor-mediated immune suppression and the blood–brain barrier pose significant challenges to the penetration and function of therapeutic immune cells [475]. In preclinical studies aiming to treat pancreatic and prostate cancer using CAR T cells targeting mucin-1 (MUC1) and prostate stem cell antigen (PSCA), low expression of the target antigens on the tumor cells led to tumor escape and a failure of the CAR T cells to eradicate the tumors [444]. Similarly, insufficient anaplastic lymphoma kinase (ALK) target density on neuroblastoma cells limited the therapeutic efficacy of ALK-specific CAR T cells [476]. Overall, various tumor-specific factors can collectively contribute to the limited efficacy of CAR T cell therapies against solid tumors, emphasizing the need for continued research efforts to develop effective strategies to overcome these barriers.

#### 3.3.3. Combinatorial Strategies for CAR T Therapy

In recent years, combination therapy has emerged as a significant method to overcome the obstacles faced by immunotherapy [477]. Combinatorial strategies include administering T cell therapies along with traditional therapies such as chemotherapy [478] and radiotherapy [479], or with other targeted therapies and immunotherapies [477].

For example, the use of small molecules to blockade the PI3K-Akt-mTOR pathway is shown to enhance in vivo expansion and functionality of CAR T cells by promoting the development of T memory stem cell (Tscm), central memory T cell (Tcm), and naïve T cell (Tn) populations, decreasing exhaustion marker expression, and increasing the ratio of CD8/CD4 T cells [477,480,481]. In patients with relapsed or refractory multiple melanomas (MMs), combination therapy with BCMA-targeted CAR T cells along with γ-secretase inhibitor (GSI), a small molecule inhibitor that effectively prevents γ-secretase-mediated cleavage of BCMA on MM cells, successfully increased BCMA surface expression on tumor cells and improved patient response rates [446].

T-cell-based therapies can also be combined with other immunotherapies to enhance their effectiveness. For instance, lenalidomide is an immunomodulatory drug that activates NK cells, heightens immune surveillance, and exhibits tumoricidal effects on MM [482]. When combined with CAR T therapy, lenalidomide enhances CAR T cell function by increasing cytotoxicity, enhancing the maintenance of memory T cell populations, and promoting the formation of immune synapses [477,483,484].

The combination of CAR T cell therapy with immune checkpoint inhibitors is also a field of great interest. Notably, the administration of monoclonal antibodies against PD-1/PD-L1, such as pembrolizumab, nivolumab, and atezolizumab, has improved the therapeutic efficiency of CAR T cell therapy for certain patient populations [477,485,486]. The combination of CAR T therapy with other immune checkpoint inhibitors, such as monoclonal antibodies against CTLA-4 (NCT00586391) and TIGIT [477] is also being explored. The use of combination therapy has emerged as a key strategy to tackle various obstacles faced in the field of immunotherapy. Extensive investigation into novel combinational strategies is urgently needed to achieve greater clinical therapeutic success.

#### 3.3.4. Multidimensional Omics Data Analyses in the Advancement of T Cell Immunotherapy

Recent advances in omics technologies have generated a vast amount of data and information. Analysis of multidimensional data encompassing genomics, epigenomics, transcriptomics, T cell receptor repertoire profiling, proteomics, metabolomics, and microbiomics is a powerful approach and can be the key to overcoming the current obstacles associated with T cell immunotherapy [487]. Indeed, the integration of multi-omic data analyses provides researchers with the ability to more accurately identify targets for the development of multiple-antigen-targeting CAR T cells that can minimize on-target, off-tumor toxicities, antigen escape, and therapy resistance [487,488,489,490,491]. For instance, combining multiple CAR targets utilizing Boolean “AND” and/or “AND-NOT” logic gating improves tumor-targeting specificity and reduces toxicity [492,493], an approach made possible through the use of comparative analyses of genomics, transcriptomics, and proteomics data from both tumor and non-malignant tissue samples to identify ideal targets [489,491].

Moreover, the multidimensional data generated by omics analyses can help researchers identify the expression of specific transcription factors, distinct epigenetic signatures, and unique metabolic states that are correlated with the enhanced therapeutic effectiveness of CAR T cells, therefore providing valuable insights into the underlying mechanisms driving CAR T cell efficacy and persistence [133,487,494,495,496]. Multi-omics data can also be used to identify tumor cell characteristics that influence response to CAR T cell therapy, features of the tumor microenvironment that can hinder CAR T cell efficacy, and potential associations between CAR T cell efficacy and the gut microbiota and microbial metabolites [487]. Undoubtedly, the leveraging of multi-omics data analyses significantly facilitates the development of novel methods to improve CAR T therapy, making it an increasingly significant research methodology.

#### 3.3.5. Advanced Methods for Monitoring CAR T Cell Potency in Patients: Flow Cytometry, CyTOF, and ImmunoPET

Non-invasive, longitudinal monitoring of CAR T cell kinetics in patients following CAR T cell infusion can provide important insights into the efficacy and activity of the therapy. Various flow cytometry assays are one of the most common methods to monitor CAR T cell populations in patients. For example, Blache and colleagues developed a 13-color/15-parameter flow cytometry panel that can characterize immune cell subpopulations in patients’ peripheral blood during CAR T cell treatment [497].

A novel approach for monitoring CAR T cell potency involves the application of cytometry by time of flight (CyTOF), which can be used to analyze trafficking and functional protein expression in CD19 CAR T cells across various patient samples and correlate them with T cell phenotypes [498]. Comparison between non-engineered T cells derived from leukapheresis, CAR T cells pre-infusion, and CAR T cells post-infection derived from peripheral blood, bone marrow, or cerebrospinal fluid of patients, revealed disparate expression profiles, indicating the spatiotemporal plasticity of CAR T cells in patients [498].

Another method for monitoring the potency of CAR T cells in patients is immune positron emission tomography (immunoPET), an imaging technique that combines the targeting specificity of mAbs with the high sensitivity of PET, enabling the visualization of the in vivo dynamics of CAR T cells [499]. ImmunoPET was successfully applied to noninvasively visualize CAR T cells by Simonetta and colleagues, who identified inducible co-stimulator (ICOS), a molecule primarily expressed on activated T cells, as highly expressed in CAR+ T cells compared to non-genetically modified CAR– T cells [500]. This expression disparity between CAR+ and CAR– T cells in patients underlined the suitability of ICOS as a target for immunoPET [499].

## 4. Conclusions

In recent years, the advent of immunotherapy has marked a paradigm shift in the treatment of cancer. This innovative approach has facilitated the development of a range of cutting-edge immunotherapies that have significantly improved the longevity of cancer patients. However, despite the promising advances and growing success of cancer immunotherapy, challenges still exist in the field. One such challenge is antigen escape due to the loss or mutation of the target antigen, which can be addressed through the use of pharmaceutical strategies that upregulate the expression of cell-surface target antigens on cancer cells. Additionally, the heterogenous nature of most cancers makes single-antigen-targeting strategies suboptimal in many cases, leading to the development of innovative CAR T therapies designed to target multiple antigens. Furthermore, poor T cell trafficking and persistence contribute to the lack of success of CAR T cells against solid tumors. Addressing this challenge can be achieved through the local administration of T cells directly into the tumor site or the use of various strategies, such as using cytokines, to increase T cell persistence. Overall, cancer immunotherapy remains a promising field with significant potential to revolutionize cancer treatment, but further research is necessary to overcome the remaining challenges. In this review, we explored the mechanisms behind the challenges faced in the field of immunotherapy, including T cell exhaustion and the immunosuppressive nature of the TME. We also surveyed the therapeutic strategies utilized to combat cancer, with a focus on highlighting noteworthy and innovative emerging immunotherapies. We hope that this review will provide a useful resource for current and future professionals working in the field of cancer immunotherapy and related areas, enabling them to discover novel targets and create optimal and efficacious targeted treatments for cancer patients.

## Figures and Tables

**Figure 1 genes-14-01008-f001:**
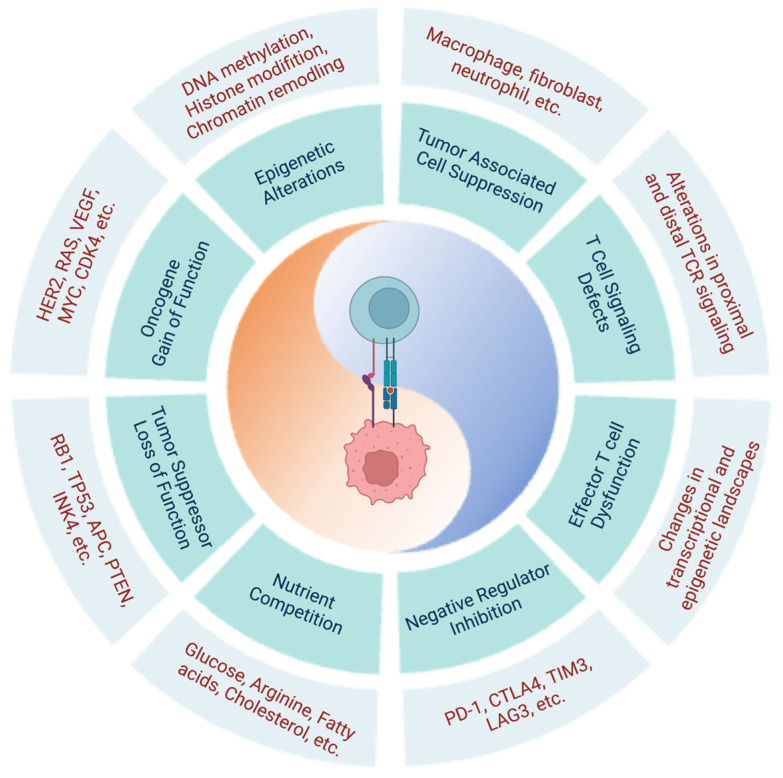
Endogenous T cell dysfunction and cancer-cell-intrinsic alterations contribute to cancer progression and failure of immunotherapy. A dynamic balance exists between tumor cells and T cells—T cells work to prevent tumors from growing and spreading, while tumor cells proliferate and metastasize, attempting to surmount immune attack. Cancer progresses when this homeostasis is broken, and tumor cell growth prevails. This figure illustrates the two distinct sets of factors that play a role in cancer progression—endogenous T cell dysfunction, illustrated by the eight panels on the right side of the figure, and cancer-cell-intrinsic alterations, depicted by the eight panels on the left side of the figure.

**Figure 2 genes-14-01008-f002:**
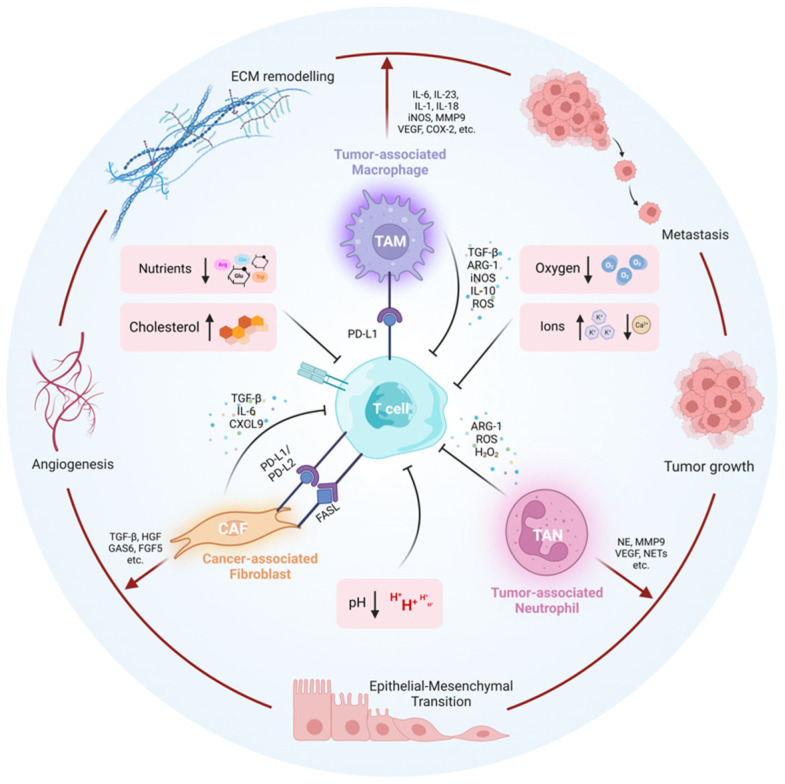
The dual contribution of tumor-associated cells to both cancer progression and T cell dysfunction. Numerous factors present in the TME play a role in driving cancer progression. Tumor-associated cells, such as TAMs, TANs, and CAFs, secrete cytokines and other factors, including TGF-β, IL-6, and ARG-1, which not only inhibit T cell function but also act on cancer cells to promote tumor growth, metastasis, angiogenesis, EMT, and ECM remodeling. Additionally, physical characteristics of the TME, such as depleted nutrients, oxygen, and essential ions, as well as elevated levels of metabolites such as ROS, cholesterol, and lactic acid, further contribute to T cell dysfunction while favoring tumor growth.

**Figure 3 genes-14-01008-f003:**
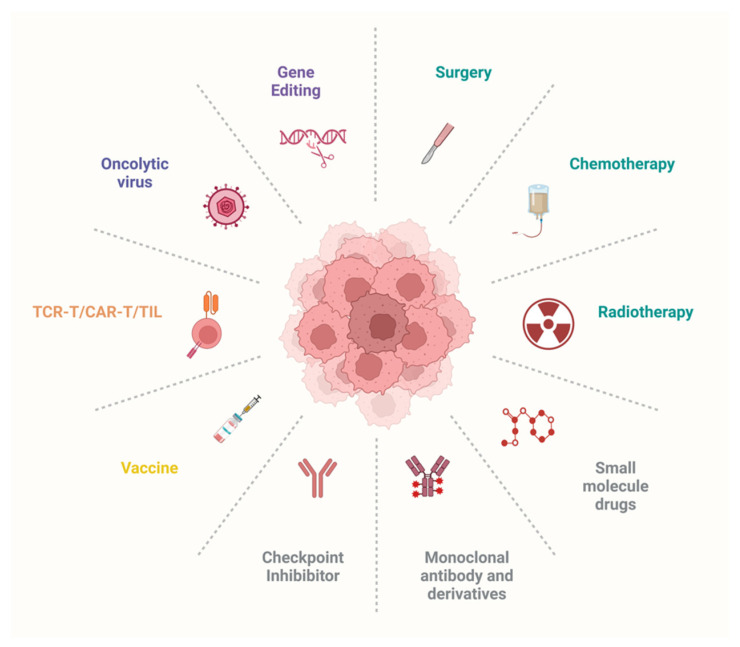
The landscape of cancer therapeutics. This figure illustrates the various therapeutics that are currently in use for the treatment of cancer. These include traditional therapies such as surgery, chemotherapy, and radiotherapy, targeted therapies including small molecule drugs, monoclonal antibodies and derivatives, oncolytic viruses, and gene editing, as well as immunotherapies including cancer vaccines, checkpoint inhibitors, and engineered T cell therapies.

**Table 1 genes-14-01008-t001:** Current ongoing or completed clinical trials of CAR T and TCR T therapies *.

NCT ID	Official Study Title	Target Antigen	Trial Phase	Targeted Conditions	Related Publications
Clinical trials of CAR T therapies					
NCT04030195	A Phase 1/2a, Open-label, Dose-escalation, Dose-expansion, Parallel Assignment Study to Evaluate the Safety and Clinical Activity of PBCAR20A in Subjects with Relapsed/Refractory (r/r) Non-Hodgkin Lymphoma (NHL), or r/r Chronic Lymphocytic Leukemia (CLL), or Small Lymphocytic Lymphoma (SLL)	Cluster of differentiation 20 (CD20)	Phase I/IIa	Non-Hodgkin lymphoma, chronic lymphocytic leukemia, small lymphocytic lymphoma	
NCT03958656	A Phase I Clinical Trial of T-cells Expressing an Anti-SLAMF7 CAR for Treating Multiple Myeloma	Signaling lymphocytic activation molecule F7 (SLAM7)	Phase I	Multiple myeloma	
NCT03338972	A Phase I Study of Adoptive Immunotherapy for Advanced B Cell Maturation Antigen (BCMA)+ Multiple Myeloma with Autologous CD4+ and CD8+ T Cells Engineered to Express a BCMA-Specific Chimeric Antigen Receptor	BCMA	Phase I	Multiple myeloma	
NCT03289455	A Single-Arm, Open-Label, Multi-Centre, Phase I/II Study Evaluating the Safety and Clinical Activity Of AUTO3, a CAR T Cell Treatment Targeting CD19 And CD22 in Pediatric And Young Adult Patients With Relapsed or Refractory B Cell Acute Lymphoblastic Leukemia	CD19, CD22	Phase I/II	B cell acute lymphoblastic leukemias	
NCT03049449	Anti-CD30 CAR T Cells with Fully-human Binding Domains for Treating CD30-expressing Lymphomas Including Anaplastic Large Cell Lymphomas	CD30	Phase 1	Lymphomas	
NCT03019055	Phase 1/1b Study of Redirected Autologous T Cells Engineered to Contain an Anti CD19 and Anti CD20 scFv Coupled to CD3ζ and 4-1BB Signaling Domains in Patients with Relapsed and/or Refractory CD19 or CD20 Positive B Cell Malignancies	CD19, CD20	Phase I/Ib	Lymphoma, non-Hodgkin lymphoma, B cell chronic lymphocytic leukemia, small lymphocytic lymphoma	
NCT02761915	A Cancer Research UK Phase I Trial of Anti-GD2 Chimeric Antigen Receptor (CAR) Transduced T-cells (1RG-CART) in Patients with Relapsed or Refractory Neuroblastoma	disialoganglioside (GD2)	Phase I	Neuroblastoma	[14]
NCT02659943	T Cells Expressing a Fully-Human Anti-CD19 Chimeric Antigen Receptor for Treating B cell Malignancies	CD19	Phase I	B cell lymphoma, non-Hodgkin’s lymphoma	[15]
NCT02215967	A Phase I Clinical Trial of T-Cells Targeting B Cell Maturation Antigen for Previously Treated Multiple Myeloma	BCMA	Phase I	Multiple myeloma	[16,17,18]
NCT02030847	Phase II Study of Redirected Autologous T Cells Engineered to Contain Anti-CD19 Attached to TCR and 4-1BB Signaling Domains in Patients with Chemotherapy Resistant or Refractory Acute Lymphoblastic Leukemia	CD19	Phase II	B cell acute lymphoblastic leukemias	[19]
NCT01583686	Phase I/II Study of Metastatic Cancer Using Lymphodepleting Conditioning Followed by Infusion of Anti-mesothelin Gene Engineered Lymphocytes	Mesothelin	Phase I/II	Cervical, pancreatic, ovarian, lung cancers; mesothelioma	[20,21]
NCT00924287	Phase I/II Study of Metastatic Cancer That Expresses Her-2 Using Lymphodepleting Conditioning Followed by Infusion of Anti-Her-2 Gene Engineered Lymphocytes	HER-2	Phase I/II	Metastatic cancers	[22,23]
NCT01460901	Phase I Study of Donor Derived, Gene Modified, Multi-virus-specific, Cytotoxic T-Lymphocytes Redirected to GD2 for Relapsed/Refractory Neuroblastoma Post-allo Stem Cell Transplantation with Submyeloblative Conditioning	disialoganglioside (GD2)	Phase I	Neuroblastoma	
NCT01454596	A Phase I/II Study of the Safety and Feasibility of Administering T Cells Expressing Anti-EGFRvIII Chimeric Antigen Receptor to Patients with Malignant Gliomas Expressing EGFRvIII	Epidermal growth factor receptor variant III (EGFRvIII)	Phase I/II	Malignant glioma, glioblastoma, brain cancer, gliosarcoma	
NCT01029366	Pilot Study of Redirected Autologous T-cells Engineered to Contain Anti-CD19 Attached to TCR and 4-1BB Signaling Domains in Patient with Chemotherapy Resistant or Refractory CD19+ Leukemia and Lymphoma	CD19	Phase I	Leukemias, lymphomas	[19,24,25]
Clinical trials of TCR T therapies					
NCT02992743	A Pilot Study of NY-ESO-1c259T Cells in Subjects with Advanced Myxoid/Round Cell Liposarcoma	New York esophageal antigen-1 (NY-ESO-1)	Phase II	Liposarcoma	
NCT02650986	A Phase I/IIa Study of TGFß Blockade in TCR-Engineered T Cell Cancer Immunotherapy in Patients with Advanced Malignancies	NY-ESO-1	Phase I/IIa	Various carcinomas, neoplasms, sarcomas	
NCT02588612	A Pilot Open-Label Clinical Trial Evaluating the Safety and Efficacy of Autologous T Cells Expressing Enhanced TCRs Specific for NY-ESO-1 in Subjects with Stage IIIb or Stage IV Non-Small Cell Lung Cancer (NSCLC)	NY-ESO-1	Phase I	Non-small cell lung cancer	
NCT02280811	A Phase I/II Study of T Cell Receptor Gene Therapy Targeting HPV-16 E6 for HPV-Associated Cancers	HPV-16 E6	Phase I/II	(HPV)-16+ cancers (cervical, vulvar, vaginal, penile, anal, and oropharyngeal cancers)	[26,27,28]
NCT02111850	A Phase I/II Study of the Treatment of Metastatic Cancer That Expresses MAGE-A3 Using Lymphodepleting Conditioning Followed by Infusion of HLA-DP0401/0402 Restricted Anti-MAGE-A3 TCR-Gene Engineered Lymphocytes and Aldesleukin	Melanoma antigen family A, 3 (MAGE-A3)	Phase I/II	Cervical, renal, urothelial, breast cancers; melanoma	
NCT04015336	A Phase II Study of E7 TCR T Cell Induction Immunotherapy for Stage II and Stage III HPV-Associated Oropharyngeal Cancer	E7 (HPV oncoprotein)	Phase II	Papillomavirus infections, oropharyngeal neoplasms	
NCT01567891	A Phase I/IIa, Open Label Clinical Trial Evaluating the Safety and Efficacy of Autologous T Cells Expressing Enhanced TCRs Specific for NY-ESO-1 in Patients with Recurrent or Treatment Refractory Ovarian Cancer.	NY-ESO-1	Phase I/IIa	Ovarian cancer	
NCT00393029	Phase II Study of Metastatic Cancer That Overexpresses p53 Using Lymphodepleting Conditioning Followed by Infusion of Anti-p53 T Cell Receptor (TCR)-Gene Engineered Lymphocytes	Protein 53 (p53)	Phase II	Metastatic cancers overexpressing p53	
NCT01343043	A Pilot Study of Genetically Engineered NY-ESO-1 Specific NY-ESO-1ᶜ^2^⁵⁹T in HLA-A2+ Patients with Synovial Sarcoma (NY-ESO-1)	NY-ESO-1	Phase I	Neoplasms	[29,30,31]
NCT00509288	Phase II Study of Metastatic Melanoma Using Lymphodepleting Conditioning Followed by Infusion of Anti-MART-1 F5 TCR-Gene Engineered Lymphocytes	Melanoma antigen recognized by T-cells (MART-1)	Phase II	Melanoma, skin cancer	[32,33,34,35]

* Updated April 2023.

**Table 2 genes-14-01008-t002:** Alterations in negative regulator expression and changes in the transcriptional and epigenetic landscape of T cells contribute to immune dysfunction.

		Molecule	Alteration	Consequence	Associated Malignancy	References
Negative regulators	Cell Surface	PD-1	Overexpressed in T cells upon chronic antigen exposure; denotes exhausted phenotype	Limited T cell survival and function; associated with T cell dysfunction	Oncogenesis	[11,12,48,51,56,57,58,63,81,88,89,90,91,92,93,94,95,96,97,99,128,130,146]
		TIM-3				
		LAG-3				
		CTLA-4				
		BTLA				
		TIGIT				
		CD244				
		CD160				
	Intracellular	SHP-1	Decreased expression of SHP-1	Hyperactive TYK2 and JAK1 kinases	Lymphomagenesis	[106,107]
		SHP-2	Lower levels of phosphorylated SHP-2	Metastasis; increased release of inflammatory cytokines; accumulation of MDSCs	Melanoma (murine model)	[108]
Transcription & epigenetic factors		DUSP2 (PAC1)	Highly expressed in TILs	Recruitment of Mi-2β nucleosome-remodeling and histone-deacetylase complex; facilitated T cell exhaustion and loss of proliferative and effector functions	Various cancers	[134,135]
		Blimp-1	Overexpressed in exhausted CD8+ T cells	Increased expression of inhibitory receptors; repression of memory CD8+ T cell differentiation	Various cancers	[54,84,128,131,132,146,147]
		EOMES		Directly controls expression of TIM-3; antagonization of T-bet		
		BATF		Expression of inhibitory receptors; repression of TCF1, a transcription factor required for memory T cell differentiation		
		T-bet	Decreased expression in dysfunctional T cells	T-bet directly represses expression of PD1	Various cancers	[54,128,129]
		TOX	Critical for CD8+ T cell exhaustion	Required for gene expression of inhibitory receptors (e.g., Pdcd1, Entpd1, Havcr2, Cd244, and Tigit)	Various cancers	[54,55,57,98,118,119,120,121]
		NR4A1	Expressed at high levels in tolerant T cells		Various cancers	[110,112,113,114,115,116]
			Critical for development of dysfunctional exhausted T cells	Attenuated T cell effector functions and upregulation of inhibitory receptors		[54,55,57,98,118,119,120,121]
		TET2	Mutations	Heightened hypermethylation of T cell signaling/differentiation genes	Lymphomas	[75,77,78,143,144,145]
		DNMT3A				
		IDH2				

**Table 3 genes-14-01008-t003:** Key molecular mechanisms of oncogenesis in tumor cells.

		Gene	Tumor Type	Reference
Genetics	Oncogenes	BCL2	B cell leukemia, lymphoma	[148]
		BCR/ABL	Leukemia	[149,150]
		BRAF	Melanoma, papillary thyroid carcinoma	[151]
		CDK4	Various tumors	[152]
		C-KIT	Acute myeloid leukemia, myelodysplastic syndrome (MDS), MDS-derived AML	[153]
		EGFR	Lung tumor	[154,155]
		ERBA	Breast cancer	[156]
		FGF5	Hepatocellular carcinoma, colorectal cancer, prostate cancer	[157,158]
		FMS	Feline McDonough sarcoma	[159]
		FOS	HNSCC, breast cancer, various tumors	[160]
		GLI	Various tumors	[161]
		HER2	Breast tumor	[162]
		HST	Esophageal cancer, stomach cancer	[163]
		JUN	Various tumors	[164]
		MDM2	Various tumors	[165]
		MET	Hereditary renal papillary carcinoma	[166]
		MYC	Various tumors	[167]
		RAS	Various tumors	[168,169,170,171]
		RET	Papillary thyroid carcinoma	[172]
		SIS	Breast tumor	[173]
		SOS	Ras-related cancer	[174]
		SRC	Various tumors	[175]
		TTG	Various tumors	[176]
		VEGF	Angiogenesis	[177]
		VEGFR	Migration of cancer cells	[178,179]
	Tumor suppressors	APC	Colon/rectum carcinoma, adenomatous polyposis	[180,181]
		ARF (p14)	Melanoma	[181]
		BRCA1, BRCA2	Breast, ovarian, pancreatic carcinomas	[180,181,182,183]
		CDH1	Gastric cancer	[181]
		CHK 1/2	Li-Fraumeni syndrome	[181]
		DCC	Colorectal carcinoma	[181]
		DPC4	Pancreatic carcinoma	[180]
		INK4 (p16)	Melanoma, lung carcinoma, brain tumors, leukemia, lymphoma	[180,181]
		MADR2	Colon/rectum carcinoma	[180]
		MLH1, MSH2, MSH6	Colorectal cancer	[181]
		NF1	Neurofibrosarcoma, neurofibromatosis type I	[180,181]
		NF2	Meningioma	[180]
		PTC	Basal cell carcinoma	[180]
		PTEN	Brain tumors; melanoma; prostate, endometrial, kidney, lung carcinomas	[180]
		RB1	Retinoblastoma	[184]
		TP53	Brain tumors; breast, colon/rectum, esophageal, liver, lung carcinomas; sarcomas; leukemias, lymphomas, Li-Fraumeni syndrome	[180,181]
		VHL	Renal cell carcinoma	[180]
		WT1, WT2	Wilms’ tumor	[180,185]
Epigenetics	DNA methylation	AID	Chronic myeloid leukemia	[186]
		DNMT1	Colorectal, non-small-cell lung, pancreatic, gastric, breast cancer	[187,188]
		DNMT3A	Myelodysplastic syndromes, acute myeloid leukemia	[189,190,191]
		DNMT3B	ICF syndrome, SNPs in breast and lung adenoma	[192,193]
		IDH1/2	Glioma, acute myeloid leukemia	[194,195,196]
		MADR2	Colon/rectum carcinoma	[180]
		MBD1/2	Lung cancer, breast cancer	[197]
		MLH1, MSH2, MSH6	Colorectal cancer	[181]
		MLL1/2/3	Bladder TCC, hematopoietic, non-Hodgkin lymphoma, B cell lymphoma, prostate cancer	[198,199]
	Histone modifiers	BMI-1	Ovarian, mantle cell lymphomas, Merkel cell carcinomas	[200,201]
		CREBBP (CBP/KAT3A)	Gastric and colorectal, epithelial, ovarian, lung, esophageal cancer	[202]
		EP300 (P300/KAT3B)	Breast, colorectal, pancreatic cancer	[202]
		EZH2	Breast, prostate, bladder, colon, pancreas, liver, gastric, uterine tumors, melanoma, lymphoma, myeloma, Ewing’s sarcoma	[203,204]
		G9a	HCC, cervical, uterine, ovarian, breast cancer	[205]
		HDAC2	Colonic, gastric, endometrial cancer	[206]
		JARID1B/C (KDM5C)	Testicular cancer, breast cancer, RCCC	[207]
		LSD1	Prostate	[207]
		PCAF	Epithelial	[202]
		PRMT1/5	Breast, gastric cancer	[202]
		SIRT1, HDAC5/7A	Breast, colorectal, prostate cancer	[202]
		UTX (KDM6A)	Bladder, breast, kidney, lung, pancreas, esophagus, colon, uterus, brain, hematological malignancies	[207]
	Chromatin remodelers	ARID1A (BAF250A)	Ovarian clear cell carcinomas, endometrioid carcinomas, endometrial carcinomas	[208,209]
		ARID2 (BAF200)	Primary pancreatic adenocarcinomas	[210]
		BRD7	Bladder TCC	[211]
		BRM (SMARCA2)	Prostate, basal cell carcinoma	[212,213]
		CHD4/5	Colorectal and gastric cancer, ovarian, prostate, neuroblastoma, hematopoietic	[214,215,216]
		CHD7	Gastric and colorectal cancer	[217]
		P400/Tip60	Lymphoma, colon, head and neck, breast cancer	[218]
		PBRM1 (BAF180)	Breast tumors	[219]
		SNF5 (SMARCB1, INI1)	Lung, rhabdoid, medulloblastoma	[220]
		SRCAP	Prostate	[221]

## Data Availability

Not applicable.

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
