# Peer review of "The Interplay between T Cells and Cancer: The Basis of Immunotherapy"

_genes, 2023, doi:10.3390/genes14051008_

Round 1

Reviewer 1 Report

The manuscript by Chen and colleagues describes the role of T cells in cancer and in different T-cell based immunotherapies. The review is very interesting, well-organized and well-written. However, there are some issues that can be improved:

1. Authors should consider to add the paragraph or section on T cell metabolism

2. Eventhough authors have prepared very nice Figures, however the explanation under all Figures is missing. Authors should put explanation under each Figure citing the source

3. Authors should consider to add the Table of T-cell based immunotherapies in clinical trials 

4. Authors should add the information regarding the use of Heat shock proteins in cancer immunotherapy (including their use as vaccines) and their role in T cell stimulation

5. Authors may also consider to add the information or section at the end of the Review about combinatorial immunotherapies and their effect on T cells

Reviewer 2 Report

Here Chen C et. al. reviews the current literature on the immuotherapeutic interplay between cancer and T cells and discussed potential mechanisms of cancer cell adavantage and T cell dysfunction underlying the basis of limited T cell persistence and efficacy in the context of CAR T cell therapy in solid tumors.

Overall, the authors did a good job highlighting key mechanisms of tumor cell and other suppressive cells in the TME that limit the CAR T cell efficacy. The limitation of the review is that the authors did not extensively discussed potential ways to overcome these challenges, which I feel will be  really important for the readers. Also, the inclusion of the section on  therapeutic landscape of the cancer treatment seems to be corenring the focus of the review, that is T cell-cancer cell interplay in the context of immunotherapy.  

Below are my specific commnets.

1.  A wast majority of the review is focused on CAR-T cell response in the tumor microenvironment (TME). It would be useful for the reader to understand the mechanisms of CAR T cell maintenance in the recipient patients. In other words, where our current understanding stands about the survival and maintenance of transferred CAR T cells in the secondary lymphoid organs (SLO) of the host and when we say CAR T cells doesn't penetrate oe persists well enough in the solid tumors does that mean they simply die or recirculate elsewhere or just sits within a SLO, like lymph nodes.

2.  Authors clearly describe the contrasting findings about BATF and IRF4 overexpresion in CAR T cells. However, they did not provide possible reasons for these mixed results. 

3. Table-1. The nature of alterations of T-bet and TCF7 in TME CD8+ T cells needs to be clearly stated. Also, what is the consequences of TCF7 alteration? Further, what is the nature of RBPJ, ZBED2, ETV1, ID3, MAF alterations? Whether higher or reduced?

4. Is there any data available that warrants the discussion of tumor-type specific mechanisms underlying the limited therapeutic success of CAR T cells in solid tumors?

5. Section 3 seems to be loosely written and touch upon base to already existing treatment modalities, which have been extensively discussed and reviewed in the literature elsewhere. It would be a good idea to focus this section on T cell immunotherapy approach and novel possible ways to overcome key challenges that the authors discussed in section 1 and 2. This may include, but not restricted to, next generation of modifications required in CAR-T design  to achieve a higher clinical success. Another important aspect to discuss would be "target selection for CAR T design". (What logic would be needed to include more than one target to avoid immune escape?)  Further, how multi-dimensional omics and spatial transcriptomics data would help inform these decisions and overall what specific role they might play in the success of CAR-T cell therapies needs to be discussed. A short discussion on newer ways to monitor the potency of CAR-Ts would also be useful. Having such discussion would strengthen the overall quality and readability of the review.

6. The conclusion section also seems like a surface-level discussion. It would be great if authors provide some strong pointers that highlight the current challenges faced by T cell (CARs) immunotherapies and potential novel ways to address some of these key challenges. This may include (but not limited to) gene-targeted CAR T cells, next generation of CAR designs, combination of CAR T cell therapy with other engineered nanomedicines for delivery or persistence, and combination with CAR-NK's etc. 

7. I would encourage the authors to include some of the newer findings that elegantly showed the newer ways to overcome T cell dysfunction in the tumor context. (Saadey AA et al. Nat Immunol. 2023; Ghoneim HE et. al. Cell, 2027; Lontos K et al. J Immunother Cancer. 2023; Vignali PDA et al. Nat Immunol. 2023; Ford BR et al. Sci Immunol. 2022; Scharping NE et al. Nat Immunol. 2021). 

8. To provide more clarity to the readers, the directionality of the tumor growth, progression, and metastasis in the figure 2 needs to be clearly shown.    

Round 2

Reviewer 1 Report

The authors addressed all the comments 

Reviewer 2 Report

The authors have satisfactorily address my comments. The revised manuscript looks good and provide a balanced view of current state of CAR T cell therapy in cancer immunotherapy arsenal. 

- As the author's agreed in their response that they are willing to get rid of "RBPJ, ZBED2, ETV1, ID3, MAF" from the table 2 on the account of insufficient evidence, I would encourage to remove this part from the revised table. 

- Sections 3.1, 3.2, and figure 3 still deviate from a central theme of the review and may be removed to cut short the length of the manuscript.